

# Model calibration and streamflow simulations for the extreme drought event of 2018 on the Rhine River Basin using WRF-Hydro 5.2.0

Andrea L. Campoverde[1], Uwe Ehret[2], Patrick Ludwig[1], Joaquim G. Pinto[1]

[1]Institute of Meteorology and Climate Research (IMKTRO), Karlsruhe Institute of Technology (KIT), Karlsruhe, Germany
[2]Institute for Water and River Basin Management, Karlsruhe Institute of Technology (KIT), Karlsruhe, Germany

*Correspondence to*: Andrea L. Campoverde (andrea.campoverde@kit.edu)

**Abstract.** Recent drought events have significantly affected navigation through the Rhine River and the transportation of goods due to low water levels. It has become imperative to analyze the conditions in which these events occur to establish actions to develop adaptive measures and avoid monetary losses. The main focus of this study is to calibrate the hydrological model for extremely low water levels and to test its performance during the 2018 drought event in the Rhine River basin. WRF-Hydro was developed to complement the land-atmosphere interactions with the meteorological model Weather Research and Forecasting (WRF), and it has been mainly used to study flood events. In this study, we simulated the Rhine River basin's streamflow using the meteorological ERA5 reanalysis dataset as input data. The calibration period is 2016-2017, during which the influence of several parameters on the streamflow was evaluated and contrasted with the daily observed discharge values at gauging stations along the river. Land use cover and terrain slope were used to create spatially distributed parameter maps, thus avoiding the calibration process of testing a range of values, substantially reducing computational demands. During calibration, the importance of modeling realistic outflow values of Lake Constance became apparent due to its significant contribution to the upper Rhine basin. However, WRF-Hydro's lake module resulted in an overly strong dampening of streamflow. Lake Constance was, therefore, represented without the lake module, resulting in more realistic hydrographs and statistical scores. Overall, the calibration and validation process demonstrated that WRF-Hydro is capable of reproducing the variability of the discharge along the Rhine and capturing low water levels observed during the 2018 drought event. These results suggest that WRF-Hydro is a suitable model for analyzing recent and future drought events in the Rhine River basin under different climate conditions.

## 1. Introduction

The River Rhine Basin has been susceptible to extreme events in the past. Most cases are related to flood events, which have increased in southern Germany per decade from 1960-2010, based on an evaluation of observed data across Europe (Blöschl et al., 2019). On the other hand, drought events have also become prominent in recent years. A historical streamflow analysis of the Rhine River's basin from 1901 to 2010 by the International Commission for the Protection of the Rhine (ICPR) revealed





a slight increase in low water levels across the entire basin during this period. This increase in discharge levels can be attributed to water management strategies designed to mitigate the effects of water scarcity in regions outside the Rhine River, including water transfer from other basins for irrigation, water intake, and storage for hydropower in the Alps region (ICPR, 2018). However, these solutions have had negative effects on the shipping sector in the Rhine, which has been affected by low water levels. This issue has resulted in the formation of bottlenecks along the river and an increase in freight rates, therefore, a

negative economic impact on the industry sector (Kriedel, 2022). The Rhine River is the most frequent waterway transporting materials and goods from and to resident companies, tourism, and recreation in Germany (ICPR, 2018). Over 85% of the country's freight, including coal, crude oil, and other goods, is transported on the river, making navigability restrictions a significant challenge for the region's supply chains (Hartz, 2022). This issue is particularly prevalent during drought periods where its patterns and trends vary in Europe. Spinoni et al. (2015) concluded that South-Western Europe's region has shown

a significant tendency to dry settings over the past 60 years, with the increase in temperature being the main driver. Ionita & Nagavciuc (2021) stated that during the last decades, Central Europe has been inclined to drought due to anomalies in temperature and potential evapotranspiration. In Germany, specifically, extreme drought events have occurred irregularly, with only a few instances observed between 1970-2009. However, the frequency of such events affecting the entire country has increased markedly since 2010 (*Dürren 1952 - 2023 (Jährlich) - UFZ*, 2023). As a consequence, it has become imperative to

analyze the impact, short-term forecasting, and long-term scenarios of drought events on the River Rhine's discharge, which has become crucial in establishing preventive measures.

During the summer of 2018, Europe was subjected to an extreme drought event characterized by high temperatures and a lack of precipitation, resulting in low groundwater levels and streamflow values (Aalbers et al., 2023). For northern and central

Europe, 2018 represented a historical event because these conditions are foreign for these regions during summer. Furthermore, these events are expected to be more frequent in a warming world setting (Rousi et al., 2023). A report on the Rhine River's basin (IKSR, 2020) indicated that the precipitation deficit represented an approximate percentage of 45% compared to the mean values from February to November. Consequently, this significantly impacted the discharge of the Rhine and its tributaries. Despite the local occurrence of heavy precipitation, it did not result in a notable rise in water levels. These extreme

drought conditions ended with a flood event in December 2018 (IKSR, 2020). In the Alpine region, high temperatures throughout spring led to a deep melting of glaciers, resulting in an overflow of the rivers nearby. However, from August onwards, there was a notable decline in the water levels of the Alpine Rhine, Lake Constance, and the downstream regions (IKSR, 2020). BAFU (2019) indicated that this drought event resulted in a 2.7% reduction in glacier volume, and the thickness of the thaw layer reached record levels. Across the basin, goods transported along the river experienced disruptions. Notably,

the Middle Rhine from the gauge at Kaub to Andernach was affected, representing a once-in-a-100-year low water duration event. Bottlenecks forced companies to resort to small cargo boats or land transportation, intensifying traffic and increasing transportation fees. This ultimately resulted in estimated monetary losses of 2 billion Euros (IKSR, 2020).





Several hydrological models have been used to assess the capability to represent the streamflow in the River Rhine's basin, as
well as changes in the discharge and extreme events under global warming. One of the models is RHINEFLOW, a distributed
(topography, land use, soil type) water balance model that tracks the inflow and outflow of a system and has been used to
estimate discharge values of the River Rhine basin (Kwadijk & Middelkoop, 1994). Middelkoop et al. (2001) and Shabalova
et al. (2003) analyzed the effects of climate change on the discharge with RHINEFLOW by establishing alterations to
temperature and precipitation. They concluded that the evident but diverse streamflow changes are shown seasonally in
projected scenarios, with a reduction during the summer months and an increase during winter. Hurkmans et al. (2008)
compared the output of the Spatial Tools for River Basins and Environment and Analysis of Management options (STREAM)
and Variable Infiltration Capacity (VIC) models. The first is a distributed water balance model, and the second is a land surface
model, which uses climate data as forcing input to simulate land-atmosphere energy and water exchanges. The authors stated
that the advantage of STREAM is its simplicity and the ability to produce annual mean values. However, VIC had a better
representation of the annual cycle, even though it requires more input data and computational resources, limiting the ability to
run the model for long periods. VIC has also been compared to the conceptual model HBV (Hydrologiska Byrans
Vattenbalansavdelnig), which simulates runoff based on rainfall, and to SWIM (Soil and Water Integrated Model), a semi-
distributed ecohydrological model that combines water processes with vegetation processes and sediment transport (Vetter et
al., 2015). The results indicated that there was no discernable difference in the performance of the three models in simulating
streamflow. Other authors (Renner et al., 2009; Junghans et al., 2011; Meißner et al., 2022) have used HBV in the River Rhine
basin and concluded that it has a good performance not only in past periods but also in various climate scenarios and for
forecasting purposes. In the Rhine, a distributed version of HBV called *OpenStreams wflow_hbv* was selected
(Tangdamrongsub et al., 2015), which comprises three sections: precipitation and snow, soil moisture, and runoff. The authors
concluded that the simulated discharge from the model is in good agreement when hydrological parameters are calibrated and
the forcing data is local. Ehmele et al. (2022) utilized another version of HBV called HBV-IWS (Institute of Hydraulic
Engineering, University of Stuttgart), which requires temperature, precipitation, and potential evapotranspiration as input data,
and uses the Muskingum approach for channel routing (He et al., 2011). It was stated that this applied version of HBV is
capable of capturing the shape of the hydrograph for historical flood events. The Precipitation-Runoff-Evapotransporation-
Hydrotope (PREVAH), another conceptual hydrological model with hydrological response units (HRU), was used and coupled
with a climate model to present the effects of future climate scenarios on the Rhine River's discharge (Bosshard et al., 2014).
A similar approach was performed with the mesoscale Hydrological Model (mHM) and the climate data from the Inter-Sectoral
Impact Model Intercomparison Project model (ISI-MIP) to determine the changes in flood events in a warming climate (Rottler
et al., 2021). These last two examples show the importance of the exchange between land and atmosphere models, particularly
for evaluating the hydrological cycle within climate scenarios.

In recent years, models representing the fluxes between land and the atmosphere have been developed. Ning et al. (2019) stated
that climate models that do not consider hydrological processes tend to produce less accurate runoff estimates for a given


basin. They further argued integrating climate and hydrological models would be advantageous for both fields. Additionally, the authors provided an extensive description of the evolution of studies on atmosphere-hydrology models, from climate

models with parameterized hydrological processes to fully coupled models in which energy and water fluxes are exchanged between the climate and hydrology models (Ning et al., 2019). Senatore et al. (2015) coupled the hydrological model WRF-Hydro with the Weather Research and Forecasting climate model (WRF) and analyzed the representation of streamflow simulations and their two-way interaction in the Crati River Basin (Southern Italy). They concluded that by coupling the models, precipitation, soil moisture content, and runoff can be better simulated than applying only WRF. Many authors (Yucel

et al., 2015; Givati et al., 2016; Silver et al., 2017; Naabil et al., 2017; Rummler et al., 2019; Ryu et al., 2017; Somos-Valenzuela & Palmer, 2018; Xue et al., 2018; Liu et al., 2020; Arnault et al., 2021; Liu et al., 2021) since then have recognized the improvement of using the coupled WRF/WRF-Hydro model for from various basins and events. To the best of our knowledge, a two-way coupled atmosphere-hydrological model has not been tested for the River Rhine.

WRF-Hydro was developed to couple hydrological model elements with atmospheric models. However, it can also work "offline" or as a stand-alone version, meaning it models the water cycle independently of a climate model (WRF) (Gochis et al., 2013). The stan-alone version of WRF-Hydro is capable of effectively simulating hydrological processes by utilizing gridded surface meteorological forcing data as input for a multi-parameter land surface model (LSM), which then takes several hydrological variables, disaggregating them into a finer grid to estimate the surface runoff with different routing schemes, and

then aggregating values back to the LSM for the next time step (Gochis et al., 2018). Yucel et al. (2015) demonstrated that the coupled WRF/WRF-Hydro model is capable of simulating flood-related features in basins on the western side of the Black Sea Region in Turkey and that a suitable calibration process led to a good performance on modeled hydrographs. Givati et al. (2016), Silver et al. (2017), Naabil et al. (2017),  Rummler et al. (2019), Ryu et al. (2017), Liu et al. (2020), Gu et al. (2021), and Quenum et al. (2022) also focused on the model's ability to capture flood or peak flow events with WRF/WRF-Hydro in

Israel, Israel and Jordan, Ghana, Southern Germany, South Korea, Northern China, Middle China, and Republic of Benin, respectively. On the other hand, Galanaki et al. (2021) and Cerbelaud et al. (2022) aimed to test the ability of WRF-Hydro offline to simulate flood events in basins in Greece and New Caledonia. Other studies, for example, by Liu et al. (2021), Yu et al. (2023), and Lee et al. (2022), have also successfully applied WRF-Hydro on low discharge values in the Xijiang and Kaidu basins in China and watersheds in South Korea, respectively, stating that the simulated streamflow values are in good

agreement with observed streamflow during dry periods.

The objective of this study is to calibrate the WRF-Hydro model in its offline version for low water events in the Rhine River basin and mainly to reproduce the observed low streamflow values during the drought event in 2018. Based on gauges from the middle Rhine downstream, this event exhibited return periods between 35-40 years for the mean minimum discharge in

seven consecutive days and a 50-100 years return period for the maximum successive days with low flows (IKSR, 2020). Our study is organized as follows: The first section provides a concise overview of historical events and trends regarding low flow

and drought events in Europe and Germany, as well as a literature review of hydrological models and WRF-Hydro. The second section describes the characteristics of the River Rhine's basin, followed by a description of the configuration of WRF-Hydro and the calibration process. In the third section, the results of the simulated period are analyzed in terms of streamflow, soil

moisture, and temperature. Our conclusions and discussions of the results are drawn in the final section.

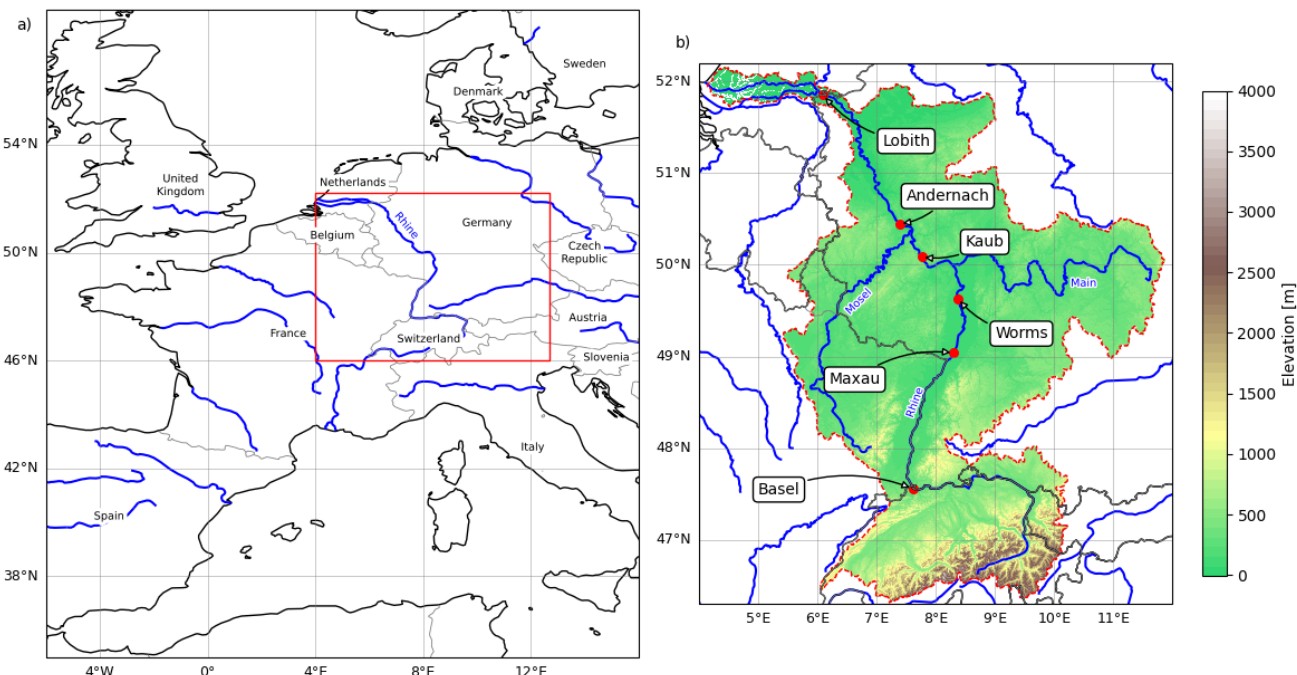

Fig 1. Rhine River's watershed geographic location (a), basin's delimitation (red dotted line), and hydrological gauge stations position (red dots) of Basel, Maxau, Worms, Kaub, Andernach, and Lobith (from upstream to downstream) (b).

**2. Materials and Methods**

**2.1 Study Area**

The Rhine River basin extends across nine countries, including Liechtenstein, Austria, Switzerland, Italy, Germany, France, Luxemburg, Belgium, and The Netherlands (Fig. 1a). The area covered by the basin is approximately 185 260 km$^2$ and is bounded between the latitudes 46.3° N and 52.2° N and longitudes 3.9° W and 12.8° E (Fig. 1b), which also defines the

boundaries for our model domain. The headwater of the Rhine is located in the Swiss Alpes, from where it flows into the North Sea in the Netherlands. Fig. 1b shows the topography of the study area, with the highest elevation situated in the Alps and the lowlands in The Netherlands. While the annual precipitation in the upper Rhine exceeds 1500 mm, the precipitation values in the middle Rhine and downstream are only between 400 and 800 mm (Pfeiffer & Ionita, 2017). The hydrological regimes of discharge of the Rhine depend on the allocation of rainfall, snow storage, and snow melt. In the upper and Alpine Rhine, the





peak values are observed during summer (Fig. 2 a,b,c) due to the melting of snow accumulated during the previous winter. The water from the Alpine region is stored in lakes, with Lake Constance being the most prominent. From the middle Rhine downstream, the regime turns pluvial, characterized by high discharge during winter (Fig. 2 d,e,f) (Pfister et al., 2004). Table 1 shows the different streamflow characteristics for mean and low discharges at the selected gauges along the Rhine River. According to the report from ICPR on the low water levels event in 2018 (IKSR, 2020) and the inventory of historical low

water conditions (ICPR, 2018), these gauges are the most affected during extreme low water level events. A comparison of the values between Basel and Lobith of the mean (MQ) and lowest mean discharge during seven consecutive days (MNM7Q) indicates that the contribution of the alpine region to the downstream streamflow is approximately 48%.

**Table 1. Observed streamflow at selected hydrological gauges along the Rhine River (see Fig. 1 for their locations) for the period 1950-2018. (MQ: Mean streamflow, MNQ: Mean low water discharge, NQ: Lowest record discharge, MNM7Q: Lowest mean discharge during seven consecutive days). Source: Global Runoff Data Center (GRDC)**

| Gauge | MQ ($m^3 s^{-1}$) | MNQ ($m^3 s^{-1}$) | NQ ($m^3 s^{-1}$) | MNM7Q ($m^3 s^{-1}$) |
|---|---|---|---|---|
| Basel | 1058 | 489 | 319 | 330 |
| Maxau | 1261 | 552 | 388 | 393 |
| Worms | 1410 | 595 | 421 | 436 |
| Kaub | 1672 | 693 | 519 | 531 |
| Andernach | 2055 | 783 | 576 | 583 |
| Lobith | 2226 | 886 | 665 | 690 |

## 2.2 Configuration of WRF-Hydro

The WRF-Hydro package was developed as a hydrological extension of the WRF model, with the objective of enabling the

coupling of the atmospheric and hydrological models. The WRF is a numerical meteorological, physically-based model and can simulate atmospheric processes based on observed data or idealized settings (NCAR, 2024). The hydrological model system WRF-Hydro can work in a two-way coupling with WRF, enabling the exchange of land-atmosphere data. Furthermore, it is possible to operate WRF-Hydro in an offline or in a one-way coupled mode, utilizing gridded meteorological forcing data (Gochis et al., 2018). The latter option is recommended for calibrating the hydrological model prior to coupling with WRF

(RafieeiNasab et al., 2022). In this chapter, we will describe the offline configuration of WRF-Hydro version 5.2.0.

The WRF-Hydro model comprises four main components. The first is the NoahMP LSM, which consists of a four-layer soil column with a total depth of 2.0 meters and layer thicknesses of 0.1, 0.3, 0.6, and 1.0 meters, which remain constant across the entire domain. The vertical fluxes of energy and water fluxes are calculated directly by the LSM. After disaggregating

variables from the LSM, a subsurface flow module calculates the lateral flow, applying a quasi-three-dimensional approach.





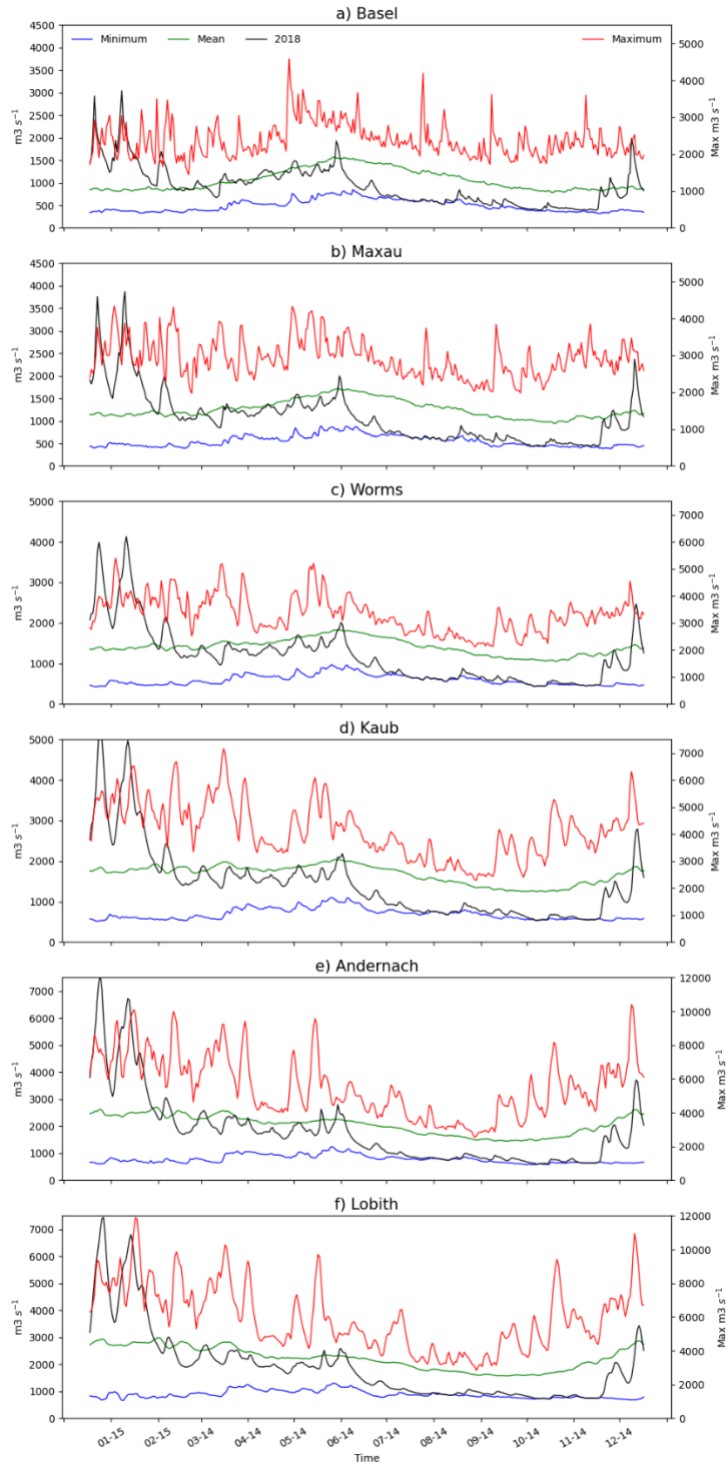

**Fig 2. Mean (green), minimum (blue), and maximum (red) daily hydrographs of selected Rhine River's Basin gauges from 1950-2018 (see Fig. 1 for their location). The hydrograph for the year 2018 is shown in black. Source: GRDC.**


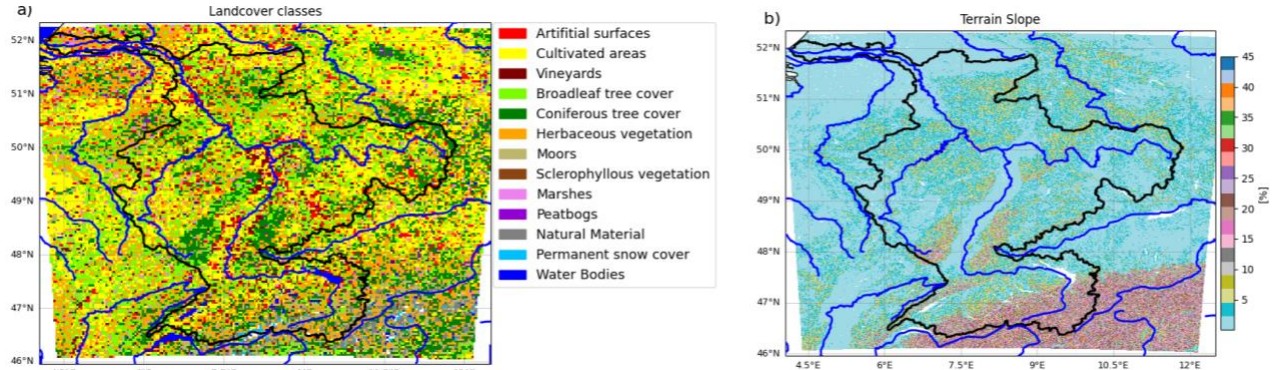

**Fig 3. Land Cover classes (a) (Malinowski, 2020) and terrain slope in percentage (b) (HydroSHEDS, 2022) for the model domain. The Rhine River's basin is delimited by a black line.**

Once the water table depth has been updated, the surface overland flow module uses diffusive wave equations for water routing
with a steepest-descent approach. Finally, the channel routing module determines the movement of water depending on the ponded water and the predefined retention depth. The routing process is based on St. Venant formulas for shallow water waves, which are used for the routing process. The WRF-Hydro model offers conceptual methods for the representation of lakes, reservoirs, and baseflow. The lakes/reservoir scheme is a level-pool system that tracks water elevation and area changes over time. The contribution of the baseflow to the streamflow is estimated using a conceptual bucket model, where the outflow is
formulated either with a pass-through or exponential bucket formulation. Both modules exert a direct influence on the discharge of the channel (Gochis et al., 2018).

The basic details of the WRF-Hydro model configuration are shown in Table 2. A digital elevation model (DEM) was acquired from HydroSHEDS (HydroSHEDS, 2022), which has a 3" (~90 m) resolution. The WRF-Hydro GIS pre-processing tool
requires this DEM and netCDF files with domain information to generate a subgrid for which the hydrological schemes will run (Sampson et al., 2021). Furthermore, the subgrid contains the channel grid, which determines the water routing across the basin and assigns the corresponding Strahler's order to all the streams. All generated channels have a trapezoidal shape, and the values of bottom width, side slope, initial water depth, and Manning's roughness coefficient are set according to channel order, as described by Gochis et al. (2018).


The model requires atmospheric forcing data, including air temperature, specific humidity, u and v wind components, surface pressure, rain rate, and incoming short and longwave radiation at a surface level. This data was obtained from the European Center for Medium-Range Weather Forecasts (ECMWF) Reanalysis version 5 (ERA5) dataset, which links observational and modeled data to create a cohesive global dataset that adheres to the fundamental principles of physics (Hersbach et al., 2020).
The temporal and spatial resolution of the input forcing data are six hours and 0.125°, respectively. The outputs of the WRF-Hydro model are aggregated to daily values. Various hydrological model parameters from the model are used in the different



schemes to estimate streamflow values throughout the channel grid. According to the literature (Senatore et al., 2015; Yucel et al., 2015; Naabil et al., 2017; Rummler et al., 2019; Ryu et al., 2017; Somos-Valenzuela & Palmer, 2018; Y. Liu et al., 2020; S. Liu et al., 2021; Yu et al., 2023) the most sensitive parameters for calibration of the WRF-Hydro are the infiltration scaling
factor (REFKDT), surface retention depth (RETDEPRTFAC), percolation (SLOPE), overflow roughness (OVROUGHRTFAC), and Manning's roughness coefficient (MANNN). In this study, these parameters were selected for calibration of the model to obtain streamflow values comparable with the observed data from the gauges listed in Table 1.

**Table 2. Configuration details for WRF-Hydro.**

|  | Structure |
|---|---|
| Noah MP grid resolution | 3000 m (144x200) |
| WRF-Hydro subgrid resolution | 375 m (1144x1592) |
| Aggregated Factor | 8 |
| Simulation period | 1st January 2016 – 31st December 2018 |
| Channel routing timestep | 60 s |
| Terrain routing timestep | 60 s |
| Channel routing option | Diffusive wave |
| Baseflow bucket model | Exponential bucket |


**Table 3. Data sources used in WRF-Hydro.**

|  | Source | Reference |
|---|---|---|
| Forcing Input Data | ERA5 Reanalysis (6h, 0.125°) | Hersbach et al., 2020 |
| DEM | HydroSHEDS – 3 "(~90 m) | HydroSHEDS, 2022 |
| Land use | USGS 24-category Land Use Categories | NCAR, 2022 |
| Observed Data | Global Runoff Dataset Center (GRDC) daily streamflow from gauges. | GRDC, 2022 |
| Land Cover | Sentinel-2 Global Land Cover | Malinowski et al., 2020 |





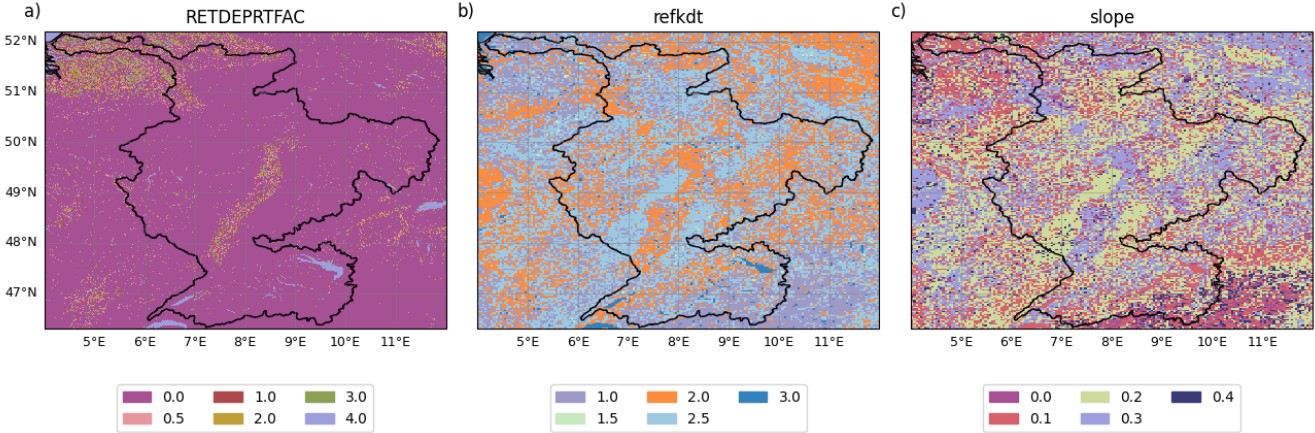

**Fig 4. Spatial distribution maps of the hydrological parameters. RETDEPRTFAC: surface retention depth (left), REFKDT:**
**infiltration scaling factor (middle), and SLOPE: percolation parameter (right).**

## 2.3 Calibration of WRF-Hydro

### 2.3.1 Hydrological parameters

The model's configuration permits a calibration process of hydrological parameters influencing the streamflow simulation. Some parameters affect the water volume, while others affect the shape of the hydrographs. The parameters that affect the
water volume are REFKDT, RETDEPRTFAC, and SLOPE. For the hydrograph shape, the sensitive parameters are MANNN and OVROUGHRTFAC. Except for MANNN, the parameters are originally set up to be constant throughout the domain. However, it has been demonstrated that a spatial distribution of calibration parameters obtained using land use information is valuable for improving the results in a complex basin (Rummler et al., 2019). The authors used information from a land cover dataset to determine the spatial distribution of the parameters REFKDT and SLOPE, resulting in acceptable simulated
discharge statistics. We adopted this approach using the data from Sentinel-2 Global Land Cover (Malinowski et al., 2020) to estimate the values of these parameters for the Rhine River basin (Fig. 4 b,c) and to accelerate the optimization process. Additionally, Yucel et al. (2015) described that the model uses the slope of the terrain to determine the value of the retention depth parameter. In regions exhibiting a slope exceeding 30%, no water accumulation occurs, whereas as the regions become flatter, the retention value increases. Using this information and transforming the DEM into the slope of the terrain, we
generated a spatially distributed map of the RETDEPRTFAC parameter (Fig. 4.a). A consensus exists among numerous authors (Yu et al., 2023; Quenum et al., 2022; Liu et al., 2021; Senatore et al., 2015; Y. Liu et al., 2020; Naabil et al., 2017; Ryu et al., 2017; Givati et al., 2016; Yucel et al., 2015) that the optimal methodology for calibrating the hydrological parameters within WRF-Hydro is a stepwise approach. The parameter value that exhibits the most favorable statistical score, indicative of its superior performance, is selected, and then the following parameter is calibrated. This procedure was eventually carried out
for the OVROUGHRTFAC and MANNN parameters using a range of values (Gochis et al., 2013; Yucel et al., 2015; Liu et al., 2020; Liu et al., 2021; Cerbelaud et al., 2022; Quenum et al., 2022; Yu et al., 2023),  as stated in Table 4.





The simulation results are compared to the daily observed data from the Global Runoff Data Center (GRDC, 2022) from the Bundesantsalt für Gewässerkunde (BfG). The selected monitoring gauge stations are listed in Table 1, and their location is

depicted in Fig. 1. Senatore et al. (2015) stated that a one-year calibration period is sufficient to assess the sensitivity of the parameters within WRF-Hydro. Given the extensive computational requirements for testing the range of the non-spatially distributed variables (OVROUGHRTFAC and MANNN), the calibration period was restricted to 2016-2017, while the validation period was set to 2018.

**Table 4. Selected hydrological parameters for the calibration process.**

| Parameter | Description | Range |
|---|---|---|
| RETDEPRTFAC | Surface retention depth | 0.0-4.0 (Fig. 4.a) |
| REFKDT | Infiltration retention factor | 1.0-2.5 (Fig. 4.b) |
| SLOPE | Percolation parameter | 0.0-0.3 (Fig. 4.c) |
| OVROUGHRTFAC | Overland flow roughness coefficient multiplier | 0.2-1.0 (0.1 increments) Gochis et al., 2013; Yucel et al., 2015; Liu et al., 2020; Liu et al., 2021; Cerbelaud et al., 2022; Quenum et al., 2022; Yu et al., 2023 |
| MANNN | Manning's roughness coefficient of the river channel | 0.2-2.0 (0.2 increments) Gochis et al., 2013; Yucel et al., 2015; Liu et al., 2020; Liu et al., 2021; Cerbelaud et al., 2022, Quenum et al., 2022; Yu et al., 2023 |

The accuracy of the model in reproducing the streamflow is assessed using standard statistical formulas, including the Nash-Sutcliffe Efficiency (NSE) (Eq. 2), Correlation (Corr) (Eq. 3), Kling-Gupta efficiency (KGE) (Eq. 4), and Bias (Eq. 5). Specifically for the evaluation of low flow (Liu et al., 2021), the Logarithmic Nash-Sutcliffe Efficiency (NSE(log)) (Eq. 6) is

used. The respective formulas are provided below.

$$NSE = 1 - \frac{\sum_{t=1}^{N}(Q_{Ot}-Q_{Mt})^2}{\sum_{t=1}^{N}(Q_{Ot}-\overline{Q_O})^2} \tag{2}$$

$$Corr = \frac{\sum_{t=1}^{N}(Q_{Mt}-\overline{Q_M})\,(Q_{Ot}-\overline{Q_O})}{\sqrt{\sum_{t=1}^{N}(Q_{Mt}-\overline{Q_M})^2 * \sum_{t=1}^{N}(Q_{Ot}-\overline{Q_O})^2}} \tag{3}$$

$$KGE = 1 - \sqrt{(Corr-1)^2 + \left(\frac{\sigma_{Q_M}}{\sigma_{Q_O}}-1\right)^2 + \left(\frac{\overline{Q_M}}{\overline{Q_O}}-1\right)^2} \tag{4}$$


$$Bias = \frac{\sum_{t=1}^{N}(Q_{Mt}-Q_{Oi})}{\sum_{i=1}^{N}Q_{Ot}} \times 100 \tag{5}$$

$$NSE(log) = 1 - \frac{log(\sum_{t=1}^{N}(Q_{Ot}-Q_{Mt})^2)}{log(\sum_{t=1}^{N}(Q_{Ot}-\overline{Q_O})^2)} \tag{6}$$





Where $t$ represents a point in time, $Q_O$ is an observed point, $Q_M$ is a simulated value, $\overline{Q_M}$ and $\overline{Q_O}$ are the are the mean values for modeled and observed values, $\sigma_{Q_M}$ and $\sigma_{Q_O}$ are the standard deviations, and N is the amount of days of the study period.


Due to its relevance to the water cycle, the soil moisture content and temperature are essential to analyzing drought events. As these variables are not part of the calibration process or input variables, we compared the simulated values with the reanalysis dataset ERA5 Land (Muñoz Sabater, 2019) as an independent check of the model realism.

### 2.3.2 WRF-Hydro's Lake Scheme

The Rhine River is divided into several sub-basins, namely the alpine, high, upper, middle, and lower Rhine. The hydrological station Basel corresponds to the alpine/high section containing Lake Constance (see Fig. 1). The upper sub-basin contains the stations Maxau and Worms. Kaub and Andernach are located in the middle Rhine, and the Lobith is situated within the lower section of the catchment. Table 1 presents the mean streamflow values at the gauges, which indicate that the streamflow at Basel (high Rhine) contributes to 75-84%, 51-63%, and 48% of the streamflow in the upper, middle, and lower Rhine,

respectively. Because of the relevant contribution of the alpine/high Rhine section on the downstream flow, the effect of Lake Constance should be considered in the modeling approach.

WRF-Hydro provides a lake scheme consisting of a level-pool routing, which follows the water elevation changes in time using parameters established with the ArcGIS Pre-Processing tool (Gochis et al., 2018). This tool uses the channel grid and a

shapefile containing the location of lakes to generate a new grid that incorporates lakes as part of the channel grid, along with files containing lake characteristics and pre-set parameters. In order to assess the ability to recreate the behavior of Lake Constance, the basin was tested without the lake module and with modifications of the lake parameters. For the latter approach, the lake outflow is estimated as follows:

$$Q_w = C_w L h^{3/2} \qquad (7)$$

where $Qw$ is the weir overflow, $Cw$ is the weir parameter, $L$ is the weir length, and $h$ is water elevation at each time step. Therefore, a range of values (0.5-0.9) with an increment of 0.1 for $Cw$, and for the weir elevation (WeirE), a range of 425m – 433m with a 2m increment was tested. After examining the lake parameters and noticing a strong dampening effect on the streamflow peaks for our area of study, we also tested the model without the lake scheme.





## 3. Results

**3.1 Streamflow calibration**

We now analyze the performance of the offline WRF-Hydro model in terms of discharge. The statistical scores of the performance of the model at each selected station of the assessment of the WRF-Hydro hydrological parameters (REFKDT, RETDEPRTFAC, SLOPE, OVROUGHRTFAC, MANNN) are presented in Table 5. The optimization of the roughness coefficients (OVROUGHFAC = 0.5, MANNN = 0.4) produced median values for NSE, KGE, CC, and absolute mean of BIAS

during the calibration period are 0.49, 0.62, 0.76, 14.18; for the validation period, they are 0.60, 0.70, and 0.88, respectively. Contrary to expectations, there is an improvement in the scores for the validation period. The reason is probably that 2016 and 2017 were wet years with high flows, resulting in higher errors. On the other hand, 2018 was a dry year, which will result in low errors. According to Knoben et al. (2019), we can assess the simulated results by comparing NSE and KGE. They stated that values are "behavioral" when NSE > 0.5 or KGE > 0.3, which is the case for the mean values of the two periods. Regarding

low water levels, NSE(log) scores for all the stations in 2018 were above 0.6. This indicates that the model can capture low water levels in an acceptable manner. The hydrographs in Fig. 5 show the simulated and observed discharge at all the gauges. Overall, there is a good agreement between the simulation and the observation data for the calibration of hydrological parameters. However, there is a slight underestimation of the streamflow upstream at Basel station and a slight overestimation downstream at Lobith station. Similar results were presented by Lui et al. (2021) for the Xijian River basin in China, which

has an area of 346.000 km$^2$. After calibration, it was concluded that the model underestimates the streamflow on the gauges in the upstream sub-basins and overestimates the discharge downstream.

**Table 5. Calibration and validation results of the WRF-Hydro's hydrological parameters at the selected stations with the lake scheme.**

| Station | Calibration (2016-2017) | | | | Validation (2018) | | | | |
|---|---|---|---|---|---|---|---|---|---|
| | NSE | KGE | CC | Bias | NSE | NSE(log) | KGE | CC | Bias |
| Basel | 0.13 | 0.30 | 0.76 | -25.28 | 0.51 | 0.65 | 0.56 | 0.75 | -10.14 |
| Maxau | 0.50 | 0.50 | 0.80 | -12.84 | 0.73 | 0.77 | 0.81 | 0.85 | 2.46 |
| Worms | 0.59 | 0.59 | 0.78 | -3.47 | 0.70 | 0.76 | 0.83 | 0.87 | 10.96 |
| Kaub | 0.53 | 0.65 | 0.75 | 7.90 | 0.62 | 0.76 | 0.73 | 0.89 | 17.44 |
| Andernach | 0.48 | 0.67 | 0.76 | 14.70 | 0.63 | 0.76 | 0.73 | 0.88 | 20.30 |
| Lobith | 0.34 | 0.68 | 0.76 | 20.88 | 0.61 | 0.75 | 0.68 | 0.90 | 22.63 |
| Median | 0.50 | 0.62 | 0.76 | 2.22 | 0.63 | 0.76 | 0.73 | 0.88 | 14.20 |


As shown in Table 5, the gauge Basel underperforms in both periods when analyzing the model's hydrological parameters. Several lake parameters (WeirE, Cw) were tested to enhance the scores, and Table 6 shows the improvement in the station Basel with the set value of WeirE = 434.1 and modified Cw = 0.5. The median values of this station for NSE, KGE, CC, and





the BIAS absolute mean during the calibration period are 0.19, 0.32, 0.73, and 13.39, respectively. However, overall, the
median values of the same formulas among all gauges are 0.46, 0.59, and 0.75 for calibration, lower than the previous setup;
in the validation period, the scores were 0.67, 0.80, and 0.85, exhibiting an improved performance.

**Table 6. Calibration and validation results of the WRF-Hydro's lake parameters at the selected stations.**

| Station | Calibration (2016-2017) | | | | Validation (2018) | | | | |
|---|---|---|---|---|---|---|---|---|---|
| | NSE | KGE | CC | Bias | NSE | NSE(log) | KGE | CC | Bias |
| Basel | 0.19 | 0.32 | 0.73 | -22.48 | 0.38 | 0.54 | 0.51 | 0.70 | -15.10 |
| Maxau | 0.49 | 0.49 | 0.77 | -11.01 | 0.66 | 0.73 | 0.75 | 0.81 | -3.50 |
| Worms | 0.57 | 0.57 | 0.76 | -1.79 | 0.70 | 0.76 | 0.83 | 0.85 | 4.77 |
| Kaub | 0.48 | 0.61 | 0.73 | 9.00 | 0.68 | 0.78 | 0.82 | 0.86 | 10.40 |
| Andernach | 0.44 | 0.63 | 0.74 | 15.38 | 0.67 | 0.78 | 0.80 | 0.85 | 12.74 |
| Lobith | 0.33 | 0.65 | 0.75 | 20.68 | 0.67 | 0.78 | 0.79 | 0.87 | 14.73 |
| Median | 0.46 | 0.59 | 0.75 | 3.61 | 0.67 | 0.77 | 0.80 | 0.85 | 7.59 |

The negative BIAS values in Basel, Maxau, and Worms (Table 5, 6) indicate that the model is reducing the contribution of
Lake Constance to the Rhine. Thus, a simulation was performed without considering the conceptual lake scheme to solve this
issue, meaning the lake is counted as part of the stream.  Table 7 contains the median values of NSE, KGE, CC, and the
absolute mean BIAS for the calibration, which are 0.28, 0.67, 0.78, and 20.77, respectively. The lower value of NSE is due to
the increase of streamflow volume in the entire basin, but KGE and CC have improved. The elevated peaks in discharge
observed during the spring period can be attributed to the simulated snowmelt. The model's estimation of the snowmelt
quantity is found to be in excess of the actual value, resulting in a significant reduction in snow depth during spring season
and a complete absence of snow by the summer months (Fig. S1). By removing the lake scheme from the model setup, Lake
Constance's contribution to the station Basel was significantly improved. Fig. 5 a, b, and c show that the lake scheme
significantly dampens the streamflow, which is corroborated when analyzing the output of the lake and also shown in Fig. 6.
Two gauges were selected for comparison: Diepoldsau (upstream) and Neuhausen (downstream). The damping effect is
noticeable in the output of the lake scheme, meaning that the model's scheme diminishes Lake Constance's contribution to the
river Rhine's discharge. Even though the scores are lower for the model without the lake scheme downstream (Kaub,
Andernach, Lobith), caused by the increase in volume to the channel, the improvement at the Basel gauge is central due to the
impact on discharge from the Lake Constance. Furthermore, the validation period yielded median scores of 0.30, 0.57, and
0.56 for NSE, KGE, and NSE(log), respectively, indicating good agreement of the model without the lake scheme to produce
low water levels. With these results in mind, it was decided that, for all further analysis, the model will not contain the lake
scheme for Lake Constance. Senatore et al. (2015) had previously observed that the lake scheme of WRF-Hydro was unable
to reproduce the behavior of the output of a dam in the Crati River (Italy), particularly during drought seasons. In the case of

Rummler et al. (2019), it was concluded that the significant positive bias downstream of Lake Walchen (Germany) during a
flood event is because, in reality, the lake is managed for hydroelectricity, which the model is not capable of reproducing.

**Table 7. Calibration and validation results of WRF-Hydro simulation without the lake scheme at the selected stations.**

| Station | Calibration (2016-2017) | | | | Validation (2018) | | | | |
|---|---|---|---|---|---|---|---|---|---|
| | NSE | KGE | CC | Bias | NSE | NSE(log) | KGE | CC | Bias |
| Basel | 0.49 | 0.75 | 0.78 | 3.78 | 0.03 | 0.36 | 0.55 | 0.68 | 11.70 |
| Maxau | 0.46 | 0.73 | 0.81 | 11.83 | 0.28 | 0.52 | 0.63 | 0.78 | 20.14 |
| Worms | 0.35 | 0.69 | 0.80 | 18.34 | 0.31 | 0.55 | 0.59 | 0.84 | 26.91 |
| Kaub | 0.11 | 0.63 | 0.78 | 25.31 | 0.29 | 0.57 | 0.54 | 0.86 | 30.98 |
| Andernach | 0.20 | 0.64 | 0.78 | 28.52 | 0.46 | 0.65 | 0.60 | 0.87 | 31.22 |
| Lobith | -0.05 | 0.58 | 0.77 | 32.00 | 0.42 | 0.62 | 0.55 | 0.89 | 32.39 |
| Median | 0.35 | 0.67 | 0.78 | 21.83 | 0.30 | 0.56 | 0.57 | 0.85 | 28.95 |

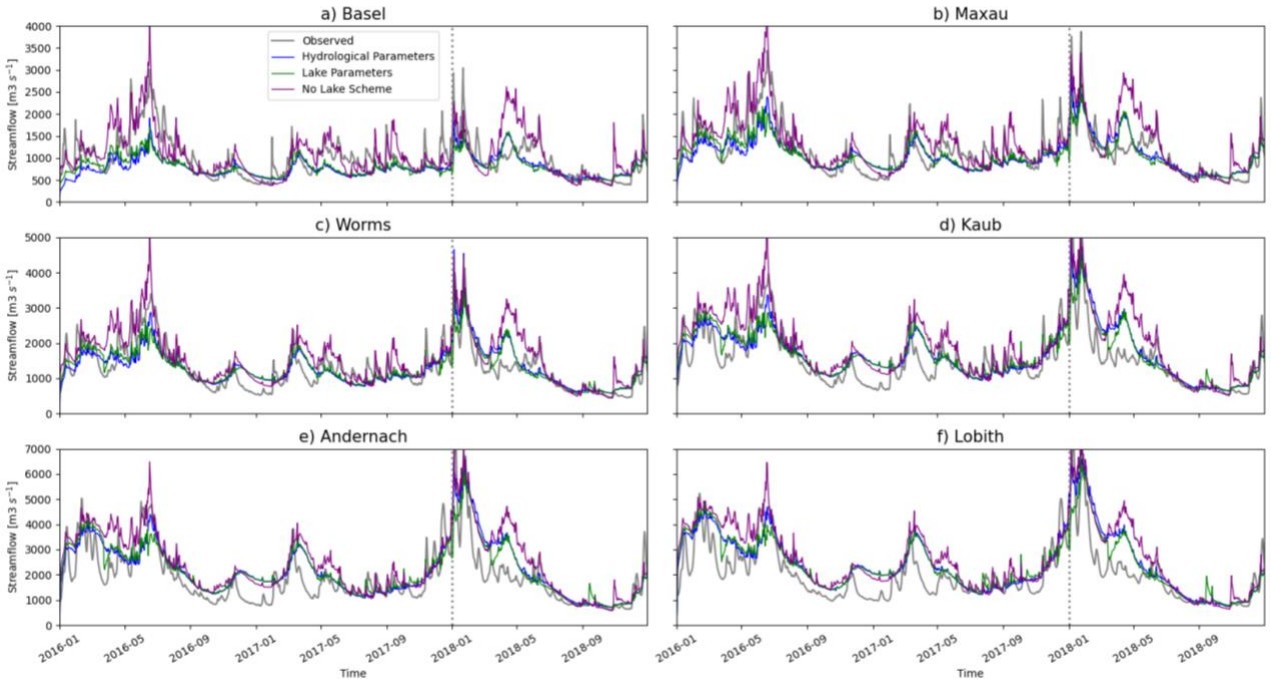

**Fig 5. Daily streamflow hydrographs after calibration of the model's hydrological parameters (blue), lake parameters (green), without the lake scheme (purple), and the observed data (gray line) from the stations (see Fig. 1 for their location). The calibration period is on the left side of the horizontal dotted line, and the validation period is on the right side.**



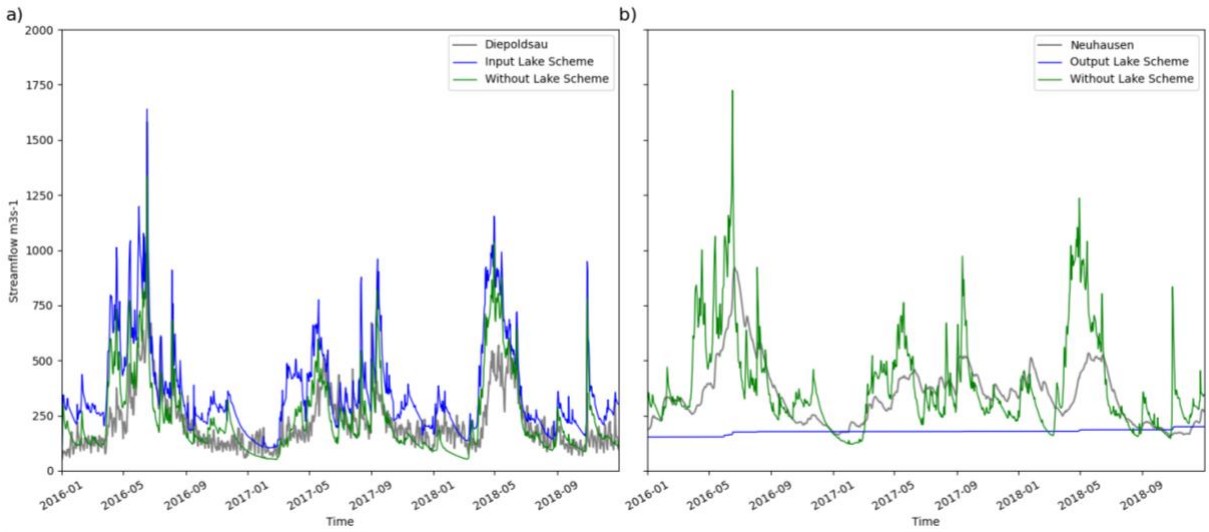

**Fig 6. Daily hydrographs at gauge Diapoldsau (upstream Lake Constance) (a) and Neuhausen (downstream Lake Constance) (b)**
**(gray), and simulated outputs of the model with (blue) and without the lake scheme (green). Source: GRDC.**

### 3.2 Soil Moisture Analysis

Soil moisture plays a significant role in shaping the duration and intensity of droughts in humid regions (van Hateren et al.,
2021). The conditions of soil moisture before and after an event can be used to specify the conditions of drought periods
(Leeper et al., 2021). Furthermore, assessing soil moisture content allows us to evaluate the performance of the model with an
uncalibrated and independent variable.

The LSM of WRF-Hydro simulates the soil moisture content at each time step at different depths, which is then used to update
the water head and, subsequently, the runoff. Due to our focus on drought analysis, we compared the soil moisture from the
third layer (50 - 150 cm) calculated with the LSM and the volumetric soil water from ERA5-Land (100 cm). It is also worth
mentioning that the LSM has a resolution of 3000 m, whereas ERA-Land has a resolution of 9000 m. Despite the discrepancy
in the depths of the variables, this comparison remains a practical one for assessing the realism of the model in terms of spatial
and temporal variability patterns, given that soil moisture was not part of the calibration process. Fig. 7a shows the spatial
distribution of the annual mean from 2016-2017 and its difference with ERA-5 Land. We observed that the model can
reproduce the spatial distribution of the soil moisture content, with minor underestimation in the Alpes area and overestimation
along the river Rhine channel and upstream of the tributary river Main.







**Fig 7. Annual mean for the period 2016-2018 (a) and the year 2018 (b) of the soil moisture content at 1-meter depth from the output of WRF-Hydro (left), ERA5-Land dataset (middle), and the difference between them (right).**

We also assessed the annual mean values for 2018 (Fig. 7b). The results show that the locations of low and high soil moisture values are similar to ERA5-Land. However, they are underestimated across the majority of the basin. Additionally, we assess the model's ability to capture the temporal variability in the entire basin (Figure 8). The correlation is 0.84, with a bias of -6.03% for spatial daily mean values. In general, the model is capable of capturing the fluctuations between peaks and low values throughout the entire period. Despite the discrepancies in soil column depth and horizontal resolution, the WRF-Hydro results are in good agreement with the ERA5-Land reanalysis dataset.

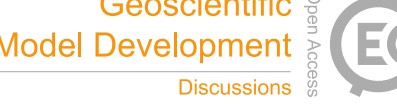

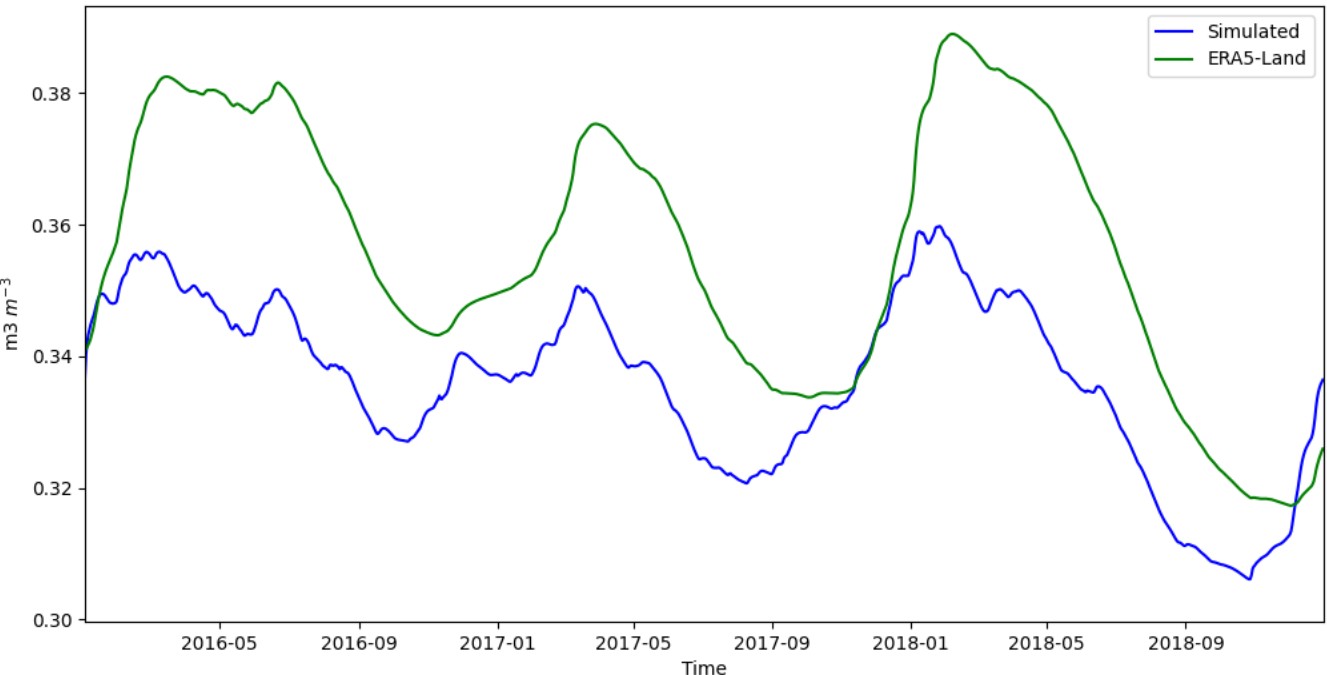

**Fig 8. Domain's spatial daily mean soil moisture content at 1-meter depth from WRF-Hydro output (blue) and ERA5-Land dataset (green).**

## 4. Conclusions

Extreme drought events can have a considerable impact on the Rhine River basin, and it is of significant importance for the industry and tourism sector to find mitigation measures for such low water level events. Therefore, the objective of this study was to demonstrate the ability of the hydrological model WRF-Hydro-offline to simulate low streamflow observed values during the 2018 drought. The calibration process of the model consisted of testing different hydrological and lake parameters, as well as switching off the lake scheme from the setup of the model because of its overly pronounced dampening effect on streamflow downstream of the lake. The statistical scores (Table 5,6) demonstrated that despite a good agreement between the simulated and observed data after calibrating the hydrological and lake parameters, the representation of the outflow of Lake Constance by the lake scheme resulted in the generation of some undesirable streamflow values. Consequently, the model was also tested without the consideration of the lake scheme. As a result, we obtained more realistic hydrographs and improved statistical scores for the comparison of simulated and observed streamflow values. The removal of the damping effect of the lake led to an increased water volume and higher peaks at all the gauges, resulting in an overestimation downstream of the Rhine River. Although this final approach did not provide the best statistical measures, it was deemed necessary to implement this configuration in order to accurately simulate the contribution of Lake Constance in case of low streamflow events in the Rhine basin. It is noteworthy that other studies have also identified shortcomings regarding the output of the scheme for lakes and reservoirs within WRF-Hydro (Senatore et al. (2015) and Rummler et al. (2019)). The results of our study indicate that



the lake scheme, in its current configuration, is not meaningful for our objectives. However, we believe that future improvements to the lake scheme, which is beyond the scope of this study, would be beneficial to be implemented for simulations in the River Rhine's catchment. The analysis of the soil moisture as an uncalibrated variable provides further

evidence of the realism of the model. A spatial and temporal variability assessment revealed that for the simulation period and during the drought year of 2018, the output of the model was in good agreement with the data from ERA5-Land, accurately capturing the peaks and low values between 2016 and 2018. For future research, the model will be applied to a climate-hydrology analysis of artificial yet realistic extreme drought events derived from long-term weather simulations for present-day simulations, as well as for future climates.


*Code and Data Availability*. The code and the exact version of WRF-Hydro (5.2.0) model as well as the dataset used in this study is freely available online and can be downloaded from Zenodo: https://doi.org/10.5281/zenodo.13221555 (Campoverde et al, 2024).The model code can also be found in https://ral.ucar.edu/projects/wrf_hydro/model-code (last access: 16 July 2024).      The      ERA5      data      are      likewise      available      from      Copernicus-ECMWF

https://cds.climate.copernicus.eu/cdsapp#!/dataset/reanalysis-era5-single-levels?tab=overview (Hersbach et al, 2023) (last access:  16  July  2024).  The  observation  data  can  be  freely  downloaded  from  Zenodo (https://doi.org/10.5281/zenodo.13221555) (Campoverde et al, 2024) and from the Global Runoff Data Center of the Bundesantsalt                          für                          Gewässerkunde                          (BfG) https://portal.grdc.bafg.de/applications/public.html?publicuser=PublicUser#dataDownload/Home (GRDC, 2022) (last access:

16 July 2023). The additional tools for the model: pre-processing ArcGIS, regrid script, and data formatting can be downloaded from Zenodo: https://doi.org/10.5281/zenodo.13221555 (Campoverde et al, 2024)

*Author contributions*. The concept of the manuscript was developed by all authors. ALC pre-processed the input for WRF-Hydro, performed simulations, analyzed data, and prepared the figures. UE contributed with methodology and data analysis.

Funding was obtained by PL and JGP. ALC wrote the initial paper draft. All authors contributed to discussions, comments, and text revisions.

*Competing interests*. The authors declare that they have no conflict of interest.

*Acknowledgements*. The spatial distribution of WRF-Hydro's hydrological parameters was done with the aid of Thomas Rummler, who provided a Python script that extracts the different parameters and transforms them into tiff files.

*Financial support*. The article processing and charges for this open-access publication were covered by the Karlsruhe Institute of Technology (KIT). ALC was partially funded by a grant from the Climate Research and the Center for Disaster Management

and Risk Reduction Technology (CEDIM). JGP and ALC thank the AXA research fund for the support.





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
