# Peer review of "Model calibration and streamflow simulations for the extreme drought event of 2018 on the Rhine River Basin using WRF-Hydro 5.2.0"

_Geoscientific Model Development, 2024_

## Referee Comment (RC1)

In this study, the authors attempt to calibrate the offline WRF-Hydro model for extremely low water levels and to test its performance during the 2018 drought event in the Rhine River basin, based on ERA5 reanalysis dataset and daily observed discharge data. The calibration process involved experimenting with various hydrological and lake parameters. Notably, the authors made the decision to disable the lake scheme due to its excessively dampening effect on streamflow downstream of the lake.

Overall I find that this manuscript present a nice piece of complete research that is clear and very well-written. The authors have demonstrated a comprehensive understanding of both the model and the complexities of hydrological dynamics in drought conditions. Their efforts to adjust the model to better reflect the observed conditions are commendable.

However, I would like to provide several comments and suggestions aimed at further enhancing the manuscript. These remarks are intended to foster constructive discussion and refinement rather than serve as critiques.

**Major comments:**

1) The abstract predominantly offers qualitative descriptions without accompanying quantitative analyses. I recommend incorporating quantitative statistical scores to enhance clarity and precision.

2) The utilization of ERA5 and ERA5-Land reanalysis datasets as both forcing and validation data for the WRF-Hydro model necessitates prior validation of their applicability within the study area. This validation is crucial to ensure the reliability and accuracy of subsequent simulations.

3) There is ambiguity regarding the estimation methodology for hydrological parameters such as REFKDT and SLOPE, derived from a land cover dataset. It would strengthen the study if the authors clarify the specific procedures used to determine these parameters.

4) The authors note unexpected model performance during calibration and validation periods. Given the focus on extreme low water events, consideration should be given to selecting a low-flow year for parameter calibration to better align with the study objectives.

5) The paper explores the impact of lake scheme variations on streamflow simulation through parameter testing and scheme deactivation. However, there lacks a detailed physical process analysis of this scheme. Furthermore, the authors should elucidate why adjustments to lake model parameters yield divergent results during calibration and validation.

6) The study attributes inaccuracies in simulated spring streamflow solely to snowmelt overestimation, neglecting to discuss other potential influences such as forcing data quality. Figure S1 indicates a consistent underestimation of snow depth throughout the simulation period, necessitating further exploration beyond seasonal dynamics.

7) While the objective of this study is to demonstrate the ability of the hydrological model WRF-Hydro-offline to simulate low streamflow observed values during the drought events, all analyses and metrics are presented for the entire year. To align with the study's focus, I recommend emphasizing discussions and analyses specific to extreme drought events in 2018.

8) Figure 7 and 8: Comparing the ERA5-Land soil moisture data with the simulation results indicates that there are obvious dry biases, especially during the low water year. However, in the previous analysis of this study, the simulation of streamflow in low water year is better than in high water years, what are the reasons for this difference? Does this mean that the model can not characterize both land surface and hydrological process parameters well?

9) Figure 8: It seems that there are some phase difference between the simulated soil moisture and ERA5-Land data. I suggest that the authors should add an explanation for these discrepancies.

**Minor comments:**

1) L135: The analysis of soil temperature can not be found in this paper.

2) L210: The term 'Slope' is identified as a soil drainage parameter; however, this is not explicitly stated in the manuscript.

3) L254: How the model's spin-up time time is set?

4) L255: There is a reference to Equation 1, which appears to be absent from the manuscript.

5) L260 and L285: The units for the variables within the equations presented on these lines are missing.

6) Line 320: Can not make sense, please rephrase it.

7) L360: It is stated that the third layer of soil moisture in the Noah-MP model is 40-100 cm. Please verify this information.

8) L370: The time period indicated in Figure 7(a) should be corrected to "2016-2017".

---

## Author Comment (AC1)

**Request:** *Major comments:*

*The abstract predominantly offers qualitative descriptions without accompanying quantitative analyses. I recommend incorporating quantitative statistical scores to enhance clarity and precision.*

**Response:**

Dear Reviewer,

We appreciate the comments provided on our manuscript. We will take your consideration in improving the abstract with more quantitative results of the calibration and validation period.

**Request:** *The utilization of ERA5 and ERA5-Land reanalysis datasets as both forcing and validation data for the WRF-Hydro model necessitates prior validation of their applicability within the study area. This validation is crucial to ensure the reliability and accuracy of subsequent simulations.*

**Response:** WRF-Hydro requires eight variables in a gridded setting as input forcing data. Therefore, ERA5 was chosen due to the open access availability of the necessary variables in a grid, and the transboundary conditions would create a complex task to obtain information from all the nine countries contained in the basin. The precipitation variable from ERA5 has been analyzed in comparison to observed data (Lavers et al. 2022). It was concluded that ERA5 is more proficient and is able to distinguish dry from wet events in Europe. Furthermore, European Center for Medium-Range Weather Forecasts datasets, such as ERA5 and ERA-Interim, have been successfully used in other studies that have used WRF-Hydro (Gu et al., 2021; Galanaki et al., 2021; Liu et al., 2021) and in the area (Rummler et al. 2018).

ERA5-Land was not used as input data for the model, and its generation is diverse from ERA5. Belsamo et al. (2009) stated that ERA5-Land uses the Hydrology Tiled ECMWF Scheme for Surface Exchanges over Land to obtain the soil moisture values and ERA5 atmospheric variables as forcing. Therefore, it is not a direct comparison with the input data to the model. We found the necessity to search for a gridded data set that will allow us to make a comparison with the results obtained from the model, which is the case of ERA5-Land.

We suggest that the distinction between the datasets, as mentioned above, will be added to the manuscript.

**Request:** *There is ambiguity regarding the estimation methodology for hydrological parameters such as REFKDT and SLOPE, derived from a land cover dataset. It would strengthen the study if the authors clarify the specific procedures used to determine these parameters.*

**Response:** The methodology followed the procedure by Rummler et al. (2018). Depending on the type of land use and its infiltration capacities, the values of REFKDT and SLOPE will differ. In areas, for example, where the impermeability is greater due to infrastructure or farming, parameter values will decrease, creating more runoff.

We suggest that the above explanation be added to the manuscript to describe the methodology in more detail.

**Request:** *The authors note unexpected model performance during calibration and validation periods. Given the focus on extreme low water events, consideration should be given to selecting a low-flow year for parameter calibration to better align with the study objectives.*

**Response:** It is acceptable to point out that a dry year could have been used during the calibration process. However, the primary objective of our study was to calibrate the model and assess its reliability to extrapolate to extreme drought conditions not seen during calibration and produce similar values of low water levels. Therefore, it is worth mentioning that despite not incorporating a dry year, the model successfully captured the extreme event.

We suggest that the explanation above be added to the manuscript.

**Request:** *The paper explores the impact of lake scheme variations on streamflow simulation through parameter testing and scheme deactivation. However, there lacks a detailed physical process analysis of this scheme. Furthermore, the authors should elucidate why adjustments to lake model parameters yield divergent results during calibration and validation*

**Response:** The contribution from the lake to the channel grid corresponds to the overflow of a weir. It consists of a mass balance, level-pool scheme that tracks the water elevation per time step using the equation in L284. The

scheme does not account for evaporation within the area of the lake or for subsurface interactions with the Land Surface Model.

The description of the lake scheme was not clear to the reader. Therefore, we will add more details on the scheme and the comparison between the output model and Lake Constance, as stated in L330-L332.

We were unsure what the Reviewer meant by "*the authors should elucidate why adjustments to lake model parameters yield divergent results during calibration and validation*". If the question is about the statistical score variations between calibration and validation, they are due to the temporal variability of the selected periods. 2016 and 2017 had the typical annual variations (peaks in winter and lower values during summer). In contrast, in 2018, the lack of precipitation for an extended period resulted in extremely low streamflow values from July to November. This explanation was pointed out in L296-L298.

**Request:** *The study attributes inaccuracies in simulated spring streamflow solely to snowmelt overestimation, neglecting to discuss other potential influences such as forcing data quality. Figure S1 indicates a consistent underestimation of snow depth throughout the simulation period, necessitating further exploration beyond seasonal dynamics.*

**Response:** Thank you for the comment. We will clarify this in the discussion. The Land Surface Model in WRF-Hydro, called Noah-MP, has multiple physics options to estimate mass and energy balance in a 1D column through parameterizations. Sthapit et al. (2022) performed a comparison analysis of the output of the LSM with reanalysis data and observed data as forcing. Their main conclusion is that because of the biases in temperature of the forcing data, there are inconsistencies with the distinction between rain and snow within the precipitation partitioning scheme of the model. Given that in our study, ERA5 was used, the conclusion also applies.

We suggest that the argument above be added to the manuscript.

**Request:** *While the objective of this study is to demonstrate the ability of the hydrological model WRF-Hydro-offline to simulate low streamflow observed values during the drought events, all analyses and metrics are presented for the entire year. To align with the study's focus, I recommend emphasizing discussions and analyses specific to extreme drought events in 2018.*

**Response:** We appreciate the comment, and we can add more detail of the drought period (July-November 2018) with the plot below in the Annex and the statistical scores on the tables in the manuscript. We will also emphasize the use of NSE(log) for performance with low streamflow values.

[Figure]

**Fig S2. Daily streamflow hydrographs of the hydrological drought period of 2018 (July-November). Model simulation is the blue line and the observed data is the gray line from the stations (see Fig. 1 for their location).**

**Table S1. Statistical analysis results of the WRF-Hydro model performance regarding the model's hydrological parameters during the hydrological drought in 2018.**

| Station | Hydrological Drought (01 July – 27 October 2018) | | | | |
|---|---|---|---|---|---|
| | NSE | NSE(log) | KGE | CC | Bias |
| Basel | 0.52 | 0.54 | 0.50 | 0.83 | -5.87 |
| Maxau | 0.69 | 0.66 | 0.71 | 0.83 | 0.36 |
| Worms | 0.64 | 0.58 | 0.78 | 0.85 | 6.29 |
| Kaub | 0.65 | 0.67 | 0.82 | 0.87 | 4.64 |
| Andernach | -0.09 | 0.22 | 0.44 | 0.88 | 12.90 |
| Lobith | -0.43 | 0.03 | 0.17 | 0.90 | 9.84 |
| Median | 0.58 | 0.56 | 0.60 | 0.86 | 5.46 |

**Table S2. Statistical analysis results of the WRF-Hydro model performance regarding the model's lake scheme parameters during the hydrological drought in 2018.**

| Station | Hydrological Drought (01 July – 27 October 2018) | | | | |
|---|---|---|---|---|---|
| | NSE | NSE(log) | KGE | CC | Bias |
| Basel | 0.25 | 0.24 | 0.49 | 0.79 | -12.21 |
| Maxau | 0.54 | 0.51 | 0.69 | 0.81 | -6.53 |
| Worms | 0.69 | 0.65 | 0.77 | 0.83 | -0.63 |
| Kaub | 0.68 | 0.65 | 0.84 | 0.86 | -2.34 |
| Andernach | 0.38 | 0.49 | 0.54 | 0.88 | 5.14 |
| Lobith | 0.06 | 0.21 | 0.26 | 0.89 | 2.76 |
| Median | 0.46 | 0.50 | 0.62 | 0.85 | -1.49 |

**Table S3. Statistical analysis results of the WRF-Hydro model performance without the model's lake scheme during the hydrological drought in 2018.**

| Station | Hydrological Drought (01 July – 27 October 2018) | | | | |
|---|---|---|---|---|---|
| | NSE | NSE(log) | KGE | CC | Bias |
| Basel | 0.24 | 0.10 | 0.70 | 0.82 | -14.20 |
| Maxau | 0.65 | 0.58 | 0.87 | 0.90 | -7.97 |
| Worms | 0.78 | 0.78 | 0.89 | 0.90 | -1.27 |
| Kaub | 0.66 | 0.66 | 0.68 | 0.91 | -0.84 |
| Andernach | -0.01 | 0.31 | 0.27 | 0.92 | 8.42 |
| Lobith | -0.54 | -0.19 | -0.05 | 0.92 | 5.26 |
| Median | 0.45 | 0.45 | 0.69 | 0.91 | -1.26 |

When considering only the low streamflow values period (July – November 2018), Fig. S2 shows that model performs these values until the end of October, as it is shown on the Tables S1, S2 and S3. The median values of the statistical scores show a good agreement with all the set ups. However, there is a better correlation and lower bias when not taking into consideration the lake scheme (Table S3) and Fig. S2 shows a better agreement with the variability of the streamflow even during this low water level period.

It is also visible that for the model there is an overestimation of the discharge from 2018-10-28 and further, with a high peak for the set up without the lake scheme (Fig. S2). This overestimation of observed streamflow is already visible at gauge Diepoldsau at the alpine Rhine, upstream of Lake Constance (Fig. S3). Therefore, the problem clearly originates from the alpine region of the basin.

[Figure]

**Fig S3. Daily streamflow hydrograph at the Diapoldsau Station**

A probable cause is that during that time, ERA5 overestimated observed precipitation in the alpine parts, and more importantly, that it overestimated air temperature, leading to falsely accounting as rain rather than snow in WRF-Hydro.

Regarding precipitation overestimation, it has been stated by other authors (e.g. Rivoire et al. 2021 or Bandhauer et al., 2022) that this is a typical phenomenon of ERA5 in the alpine regions. With respect to falsely assuming rainfall, Fig. S4 shows the spatial mean daily values of precipitation and air temperature of the alpine high-altitude region of the Rhine basin. It is noticeable that the air temperature is greater than zero for the entire period. Therefore, it is possible that WRF-Hydro considers most of the precipitation as rainfall, leading to a direct substantial rainfall-runoff event. In contrast, observations show significant new snow accumulation in the same region in order or magnitude of 0.25 – 2 m during this period (Fig. S5). The snow depth data are taken from EnviDat (Mott, R. 2023). Based on this analysis, we can conclude that the overestimation of the observed streamflow peak on October 30 is due to falsely accounting precipitation mainly as rainfall rather than snow.

[Figure]

**Fig S4. Spatial daily mean precipitation and air temperature from ERA5 for the alpine high-altitude regions of the Rhine basin.**

[Figure]

**S5. Snow depth accumulation from 2018-10-28 to 2018-11-05 (b). The location of this area is in a red square in (a).**

When studying drought conditions, we cannot omit to include the behavior of the basin before and after the event took place. Thus, it is important that our model is calibrated in accordance with a regular year and extrapolate this performance to an extended dry period. Even though the results are not perfect, we were able to find a suitable set up of the model that can achieve the basin's behavior from wet to dry events.

We suggest that the argument above be added to the manuscript as a sub chapter the result section.

**Request:** *Figure 7 and 8: Comparing the ERA5-Land soil moisture data with the simulation results indicates that there are obvious dry biases, especially during the low water year. However, in the previous analysis of this study, the simulation of streamflow in low water year is better than in high water years, what are the reasons for this difference? Does this mean that the model can not characterize both land surface and hydrological process parameters well?*

**Response:** Hydrological schemes function in a different grid and are separate from the Land Surface Model. We can calibrate different parameters that will greatly impact the volume of water in the channel grid. It was not part of our project to calibrate the soil moisture variable. The objective of the soil moisture analysis is to compare a non-calibrated variable to a data set that is the closest to observed. Our results showed that the modeled values are not out of range compared to ERA5-Land and are as good as expected, considering that differences are acceptable.

**Request:** *Figure 8: It seems that there are some phase difference between the simulated soil moisture and ERA5-Land data. I suggest that the authors should add an explanation for these discrepancies.*

**Response:** Thank you for pointing out this comment. There are differences in the timing of the peaks and valleys of soil moisture content in Fig. 8. In the manuscript (L359-L363), we stated the differences in depths between the model and ERA5-Land. Furthermore, the more significant shift occurred when the soil is reaching the lowest values per cycle. This difference could be due to the drainage of the soil moisture. The ERA5-Land model presents more volume and takes more time to drain, creating a lag when comparing the results to WRF-Hydro. The Land Surface Model of the model, on the other hand, presents a quicker response and recharges faster than the model used in ERA5-Land.

However, we don't have a satisfactory explanation for the temporal shift of high and low values changes. We would also like to point out that this is not the focus of our project, and it does not invalidate the comparison made. The results are still valid for the sanity check of an uncalibrated variable.

**Request:** *Minor comments:*

*1) L135: The analysis of soil temperature can not be found in this paper.*

**Response:** It was an overlooked error to write the temperature analysis, it is not part of our analysis and will be deleted from the manuscript.

**Request:** *2) L210: The term 'Slope' is identified as a soil drainage parameter; however, this is not explicitly stated in the manuscript.*

**Response:** In L210 it is indicated that is the percolation parameter.

**Request:** *3) L254: How the model's spin-up time time is set?*

**Response:** The spin-up time is three months. We suggest adding this detail to the manuscript.

**Request:** *4) L255: There is a reference to Equation 1, which appears to be absent from the manuscript.*

**Response:** The number of equations will be reordered in the manuscript.

**Request:** *5) L260 and L285: The units for the variables within the equations presented on these lines are missing.*

**Response:** The units of the variables will be added to the description below the equations.

*6) Line 320: Can not make sense, please rephrase it.*

**Response:** The description of the tables will be rewritten to have a better understanding.

*7) L360: It is stated that the third layer of soil moisture in the Noah-MP model is 40-100 cm. Please verify this information.*

**Response:** This value will be corrected.

*8) L370: The time period indicated in Figure 7(a) should be corrected to "2016-2017".*

**Response:** The period is correct, as stated in the manuscript. We wanted to analyze first the soil moisture results for the entire modeling period and second to focus on the drought period.

**References**

Balsamo, G., Beljaars, A., Scipal, K., Viterbo, P., van den Hurk, B., Hirschi, M., and Betts A.K.: A Revised Hydrology for the ECMWF Model: Verification from Field Site to Terrestrial Water Storage and Impact in the Integrated Forecast System, J. Hydrometeorol., 10, 623–643, https://doi.org/10.1175/2008JHM1068.1, 2009.

Bandhauer, M., Isotta, F., Lakatos, M., Lussana, C., Båserud, L., Izsák, B., Szentes, O., Tveito, O.E., and Frei, C.: Evaluation of daily precipitation analyses in e-obs (v19.0e) and era5 by comparison to regional high-resolution datasets in European regions. Int. J. Climatol., 42(2), 727–747., https://doi.org/10.1002/joc.7269, 2022

Gu, T., Chen, Y., Gao, Y., Qin, L., Wu, Y., and Wu, Y.: Improved Streamflow Forecast in a Small-Medium Sized River Basin with Coupled WRF and WRF-Hydro: Effects of Radar Data Assimilation, Remote Sens., 13, 1-23, https://doi.org/10.3390/rs13163251, 2021

Galanaki, E., Lagouvardos, K., Kotroni, V., Giannaros, T., and Giannaros, C.: Implementation of WRF-Hydro at two drainage basins in the region of Attica, Greece, for operational flood forecasting, Nat. Hazards Earth Syst. Sci., 21, 1983–2000, https://doi.org/10.5194/nhess-21-1983-2021, 2021

Lavers D.A., Simmons, A., Vamborg, F., Rodwell M.J.: An evaluation of ERA5 precipitation for climate monitoring, Q. J. R. Meteorol. Soc. 148: 3152–3165. https://doi.org/10.1002/qj.4351, 2022

Liu, S., Wang, J., Wei, J., and Wang, H.: Hydrological simulation evaluation with WRF-Hydro in a large and highly complicated watershed: The Xijiang River basin, J. Hydrol. Reg. Stud., 38, 1-20. https://doi.org/10.1016/j.ejrh.2021.100943, 2021

Mott, R.: Seasonal snow data for Switzerland OSHD - FSM2sohd. EnviDat.[data set], https://www.doi.org/10.16904/envidat.404, 2023

Rivoire, P., Martius, O., Naveau, P.: A Comparison of Moderate and Extreme ERA-5 Daily Precipitation With Two Observational Data Sets, Earth and Space Science, 8, 1-16, https://doi.org/10.1029/2020EA001633, 2021

Rummler, T., Arnault, J., Gochis, D., and Kunstmann, H.: Role of Lateral Terrestrial Water Flow on the Regional Water Cycle in a Complex Terrain Region: Investigation With a Fully Coupled Model System, J. Geophys. Res.-Atmos., 124, 507–529. https://doi.org/10.1029/2018JD029004, 2019

Sthapit, E., Lakhankar, T., Hughes, M.; Khanbilvardi, R., Cifelli, R., Mahoney, K., Currier, W.R., Viterbo, F., Rafieeinasab, A.: Evaluation of Snow and Streamflows Using Noah-MP and WRF-Hydro Models in Aroostook River Basin, Maine, Water, 14, 2145. https://doi.org/10.3390/w14142145, 2022

---

## Author Comment (AC2)

**Request:** *2018 is the target year for the study, but during the parameter calibration, 2018 was also used as the validation period. Usually, it might be better to set the validation period and the target year separately. This way, the validated model parameters can be better applied to the simulation of the target period; this is especially true if the model is used for forecasting.*

**Response:**

Dear Reviewer,

Thank you for your comments and recommendations. We were not entirely sure what the comment was here. If our assumption is correct, that the concern is that we used 2018 as both the target and calibration year. Our reply to this point is that 2018 is only consider as the validation year, and it is not used for calibration.

**Request:** *The year 2018 is the target year of the study in the paper, and an extreme drought happened in this year. In the article, the years 2016 - 2017 were used as the time for parameter calibration, but there was no extreme drought during these two years, which will inevitably lead to the calibrated parameters may not be applicable. This may also be one of the reasons why the results are not ideal. Additionally, the periods of model calibration and validation may be longer for a better and stable result.*

**Response:** It is acceptable to point out that a dry year could have been used during the calibration process. However, the primary objective of our study was to calibrate the model and assess its reliability to extrapolate to extreme drought conditions not seen during calibration and produce similar values of low water levels. The previous extreme hydrological drought was in 2003; hence, the computational requirements of the model for the Rhine basin were our main restriction. It is worth mentioning that despite not incorporating a dry year, the model successfully captured the extreme event.

We suggest that will add the explanation above to the manuscript.

**Request:** *The final simulation results do not seem ideal. Usually, an NSE result above 0.7 would be better, but in the paper, many of the results are below 0.5, and even there are some negative values. Although the hydrological situation in a larger basin is indeed more complex and difficult to simulate perfectly, is it possible to make further attempts to adjust the results better?*

**Response:** We appreciate your comment. Indeed, the results are not ideal, but they are acceptable. It was chosen to focus not only on one statistical score but on the combination of them (NSE, KGE, CC). Additionally, we had to consider all the hydrological stations, therefore using the median value. The lower values are located downstream of the basin, and this result is because there is a positive BIAS from the up-to-middle-stream of the Rhine, which contributes to the overestimation downstream, as stated in L324. Therefore, we do not see a necessity to further improve due to computational time.

We suggest that the explanation above be added to the manuscript.

**Request:** *The ERA5 data was used to drive the model operation, but it was also used as observational data to verify the model results, which is usually not recommended. It is advisable to see if there are other data, especially observational data, for verification, so that the results would be more reliable and convincing.*

**Response:** Thank you for your comment. A similar point was made by Reviewer #1. WRF-Hydro requires eight variables in a gridded setting as input forcing data. Therefore, ERA5 was chosen due to the open access availability of the necessary variables in a grid, and the transboundary conditions would create a complex task to obtain information of all the nine countries contained in the basin.

ERA5-Land was not used as input data for the model, and its generation is diverse from ERA5. Balsamo et al. (2009) stated that ERA5-Land uses the Hydrology Tiled ECMWF Scheme for Surface Exchanges over Land to obtain the soil moisture values and ERA5 atmospheric variables as forcing. Therefore, it is not a direct comparison with the input data to the model. We found the necessity to search for a gridded data set that will allow us to compare with the soil moisture results obtained from the model, which is the case of ERA5-Land. The objective was to make a sanity check with an uncalibrated variable.

We suggest that the distinction between the datasets, as mentioned above, will be added to the manuscript.

**Request:** *The title of this paper is "Model calibration and streamflow simulations for the extreme drought event of 2018 on the Rhine River Basin using WRF-Hydro 5.2.0", but there is not much analysis in the article on the simulation results of the extreme drought event in 2018. More space is devoted to the calibration of model parameters and the discussion of parameters related to the lake process. It is suggested to analyze in detail this extreme drought event and the model's simulation of this event.*

**Response:** Thank you for your comment. A similar point was made by Reviewer #1. We can add more detail of the drought period (July-November 2018) with the plot below in the Annex and the statistical scores on the tables in the manuscript. We will also emphasize the use of NSE(log) for performance with low streamflow values.

[Figure]

**Fig S2. Daily streamflow hydrographs of the hydrological drought period of 2018 (July-November). Model simulation is the blue line and the observed data is the gray line from the stations (see Fig. 1 for their location).**

**Table S1. Statistical analysis results of the WRF-Hydro model performance regarding the model's hydrological parameters during the hydrological drought in 2018.**

| Station | Hydrological Drought (01 July – 27 October 2018) | | | | |
|---------|------|----------|------|------|------|
|         | NSE  | NSE(log) | KGE  | CC   | Bias |
| Basel     | 0.52  | 0.54 | 0.50 | 0.83 | -5.87 |
| Maxau     | 0.69  | 0.66 | 0.71 | 0.83 | 0.36  |
| Worms     | 0.64  | 0.58 | 0.78 | 0.85 | 6.29  |
| Kaub      | 0.65  | 0.67 | 0.82 | 0.87 | 4.64  |
| Andernach | -0.09 | 0.22 | 0.44 | 0.88 | 12.90 |
| Lobith    | -0.43 | 0.03 | 0.17 | 0.90 | 9.84  |
| Median    | 0.58  | 0.56 | 0.60 | 0.86 | 5.46  |

**Table S2. Statistical analysis results of the WRF-Hydro model performance regarding the model's lake scheme parameters during the hydrological drought in 2018.**

| Station | Hydrological Drought (01 July – 27 October 2018) | | | | |
|---------|------|----------|------|------|--------|
|         | NSE  | NSE(log) | KGE  | CC   | Bias   |
| Basel   | 0.25 | 0.24 | 0.49 | 0.79 | -12.21 |
| Maxau   | 0.54 | 0.51 | 0.69 | 0.81 | -6.53  |
| Worms   | 0.69 | 0.65 | 0.77 | 0.83 | -0.63  |
| Kaub    | 0.68 | 0.65 | 0.84 | 0.86 | -2.34  |
| Andernach | 0.38 | 0.49 | 0.54 | 0.88 | 5.14  |
| Lobith  | 0.06 | 0.21 | 0.26 | 0.89 | 2.76   |
| Median  | 0.46 | 0.50 | 0.62 | 0.85 | -1.49  |

**Table S3. Statistical analysis results of the WRF-Hydro model performance without the model's lake scheme during the hydrological drought in 2018.**

| Station | Hydrological Drought (01 July – 27 October 2018) | | | | |
|---------|-------|----------|-------|------|--------|
|         | NSE   | NSE(log) | KGE   | CC   | Bias   |
| Basel   | 0.24  | 0.10  | 0.70  | 0.82 | -14.20 |
| Maxau   | 0.65  | 0.58  | 0.87  | 0.90 | -7.97  |
| Worms   | 0.78  | 0.78  | 0.89  | 0.90 | -1.27  |
| Kaub    | 0.66  | 0.66  | 0.68  | 0.91 | -0.84  |
| Andernach | -0.01 | 0.31 | 0.27 | 0.92 | 8.42  |
| Lobith  | -0.54 | -0.19 | -0.05 | 0.92 | 5.26   |
| Median  | 0.45  | 0.45  | 0.69  | 0.91 | -1.26  |

When considering only the low streamflow values period (July – November 2018), Fig. S2 shows that model performs these values until the end of October, as it is shown on the Tables S1, S2 and S3. The median values of the statistical scores show a good agreement with all the set ups. However, there is a better correlation and lower bias when not taking into consideration the lake scheme (Table S3) and Fig. S2 shows a better agreement with the variability of the streamflow even during this low water level period.

It is also visible that for the model there is an overestimation of the discharge from 2018-10-28 and further, with a high peak for the set up without the lake scheme (Fig. S2). This overestimation of observed streamflow is already visible at gauge Diepoldsau at the alpine Rhine, upstream of Lake Constance (Fig. S3). Therefore, the problem clearly originates from the alpine region of the basin.

[Figure]

**Fig S3. Daily streamflow hydrograph at the Diapoldsau Station**

A probable cause is that during that time, ERA5 overestimated observed precipitation in the alpine parts, and more importantly, that it overestimated air temperature, leading to falsely accounting as rain rather than snow in WRF-Hydro.

Regarding precipitation overestimation, it has been stated by other authors (e.g. Rivoire et al. 2021 or Bandhauer et al., 2022) that this is a typical phenomenon of ERA5 in the alpine regions. With respect to falsely assuming rainfall, Fig. S4 shows the spatial mean daily values of precipitation and air temperature of the alpine high-altitude region of the Rhine basin. It is noticeable that the air temperature is greater than zero for the entire period. Therefore, it is possible that WRF-Hydro considers most of the precipitation as rainfall, leading to a direct substantial rainfall-runoff event. In contrast, observations show significant new snow accumulation in the same region in order or magnitude of 0.25 – 2 m during this period (Fig. S5). The snow depth data are taken from EnviDat (Mott, R. 2023). Based on this analysis, we can conclude that the overestimation of the observed streamflow peak at October 30 is due to falsely accounting precipitation mainly as rainfall rather than snow.

[Figure]

**Fig S4. Spatial daily mean precipitation and air temperature from ERA5 for the alpine high-altitude regions of the Rhine basin.**

[Figure]

**S5. Snow depth accumulation from 2018-10-28 to 2018-11-05 (b). The location of this area is in a red square in (a).**

When studying drought conditions, we cannot omit to include the behavior of the basin before and after the event took place. Thus, it is important that our model is calibrated in accordance with a regular year and extrapolate this performance to an extended dry period. Even though the results are not perfect, we were able to find a suitable set up of the model that can achieve the basin's behavior from wet to dry events.

We suggest that the argument above be added to the manuscript as a sub chapter the result section.

**References**

Balsamo, G., Beljaars, A., Scipal, K., Viterbo, P., van den Hurk, B., Hirschi, M., and Betts A.K.: A Revised Hydrology for the ECMWF Model: Verification from Field Site to Terrestrial Water Storage and Impact in the Integrated Forecast System, J. Hydrometeorol., 10, 623–643, https://doi.org/10.1175/2008JHM1068.1, 2009.

Bandhauer, M., Isotta, F., Lakatos, M., Lussana, C., Båserud, L., Izsák, B., Szentes, O., Tveito, O.E., and Frei, C.: Evaluation of daily precipitation analyses in e-obs (v19.0e) and era5 by comparison to regional high-resolution datasets in European regions. Int. J. Climatol., 42(2), 727–747., https://doi.org/10.1002/joc.7269, 2022

Mott, R.: Seasonal snow data for Switzerland OSHD - FSM2sohd. EnviDat.[data set], https://www.doi.org/10.16904/envidat.404, 2023

Rivoire, P., Martius, O., Naveau, P.: A Comparison of Moderate and Extreme ERA-5 Daily Precipitation With Two Observational Data Sets, Earth and Space Science, 8, 1-16, https://doi.org/10.1029/2020EA001633, 2021

---

## Author Comment (AC3)

**Request**: *I have major concern related to the scientific novelty of this study. In L107, the authors motivate their study by the fact that the two-way coupled WRF/WRF-Hydro has not been tested for the River Rhine. I find this motivation too weak. Neither from methodological point of view, nor from the case study perspective I found significant advances in model development and application. The profound review presented by the authors clearly demonstrates that many other hydrological models have been previously setup and applied in the Rhine basin. Also, WRF-Hydro has been setup, calibrated and applied in many different catchments, though seemingly not in the Rhine basin. The calibration methodology seems to be based on a two-stage parameter adjustment across their plausible ranges. This sounds like a very standard procedure applied in many modelling study.*

**Response:**

Dear Reviewer,

We appreciate your comments on our manuscript and will answer them to provide a more in-depth view of our project.

It is apparent from your comment that the objectives and novel aspects of our study were not clearly formulated in the manuscript. We are focusing not only on testing the model in the Rhine basin but specifically to do so for drought conditions. As stated in the manuscript, the most common implementation of WRF-Hydro has been performed for flood events. Additionally, we include several novel aspects, like the use of spatially distributed parameters methodology using land cover (Rummler et al. 2017) for the infiltration scaling (REFKDT) and percolation factor (SLOPE). Additionally, we propose the use of the slope of the terrain from the DEM to establish different values of surface retention depth (RETDEPRTFAC) rather than uniform values throughout the basin. With this approach, we document that the methodology works for the Rhine basin and specifically for drought situations.

We suggest enhancing the current manuscript by highlighting in more detail the objectives and the novel aspects of the methodology that we are proposing, like the new approach for the evaluation of the parameter RETDEPRTFAC in a complex basin. Based on our knowledge, no publications have a similar approach to our project. In case the reviewer is aware of some studies with similar approaches, we would like you to point is to them, so that we can include them in the reasoning of the paper.

**Request:** *The fact that the lake scheme did not work properly and needs to be switched off is not deeply investigated and critically discussed. Why does it lead to poorer results? How can it be potentially improved? It is not sufficient to point out to previous studies that identified the same flaw.*

**Response:** Thank you for your comment. Finding an improvement model's lake scheme is not the objective of our project.. The model WRF-Hydro version 5.2, used in this study, has a scheme for lakes representing weir overflow. The formula in L284 gives the methodology to estimate the lake contribution to the channel grid. The equation considers a weir parameter, the length of the weir (set with the ArcGIS pre-processing tool), and water elevation at the time step. The less desired results result are the result of the increase in volume from upstream and its accumulation downstream. We have added a more in-depth description of the Lake Scheme in the comments of the other reviewers. Our suggestion for improving the scheme is the interactive lake retention approach.

We propose improving the description of the Lake Scheme from WRF-Hydro and including our suggestion for improving the scheme in the conclusions, which could be used as a starting point for another publication.

**Request:** *Overall, I am not convinced that this study presents a significant model advancement. Neither, it is clear what we can learn from the presented application. I regret to suggest the rejection of the manuscript. Maybe the authors could develop a more appealing test case for the Rhine basin and consider a journal focusing on regional studies, e.g. Journal of Hydrology: Regional Studies. Alternatively, a methodological improvement focusing on developing a more profound lake and reservoir scheme seems promising.*

**Response:** The purpose of selecting GMD as a journal and specifically modeling evaluation is because the Aims & Scope of the journal states that the type of manuscripts that they allow for peer review include "full evaluations of previously published models" which is the case for WRF-Hydro and what the content of our manuscript is. Numerous previous publications regarding WRF-Hydro have focused on flood events; we provided an approach adequate to model extreme drought in the Rhine Basin. Furthermore, we evaluated the methodology of using spatially distributed hydrological infiltration parameters using land cover (Rummler et al.

2017), and we propose the use of terrain slope to determine spatially distributed values of that water retention depth parameter, which has a direct influence on the channel routing scheme. Providing model advancements for WRF-Hydro, i.e., improvement of the lake scheme, can be a suggestion for another publication.

**References**

Rummler, T., Arnault, J., Gochis, D., and Kunstmann, H.: Role of Lateral Terrestrial Water Flow on the Regional Water Cycle in a Complex Terrain Region: Investigation With a Fully Coupled Model System, J. Geophys. Res.-Atmos., 124, 507–529. https://doi.org/10.1029/2018JD029004, 2019